# Adolescent cardiorespiratory fitness and risk of cancer in late adulthood: A nationwide sibling-controlled cohort study in Sweden

Marcel Ballin[1]*, Daniel Berglind[2,3,4], Pontus Henriksson[5], Martin Neovius[6], Anna Nordström[7,8], Francisco B. Ortega[9,10], Elina Sillanpää[10,11], Peter Nordström[1‡], Viktor H. Ahlqvist[1,12,13‡]

1 Clinical Geriatrics, Department of Public Health and Caring Sciences, Uppsala University, Uppsala, Sweden, 2 Department of Global Public Health, Karolinska Institutet, Stockholm, Sweden, 3 Centre for Epidemiology and Community Medicine, Region Stockholm, Stockholm, Sweden, 4 Center for Wellbeing, Welfare and Happiness, Stockholm School of Economics, Stockholm, Sweden, 5 Department of Health, Medicine and Caring Sciences, Linköping University, Linköping, Sweden, 6 Clinical Epidemiology Division, Department of Medicine, Karolinska Institutet, Stockholm, Sweden, 7 Rehabilitation Medicine, Department of Medical Sciences, Uppsala University, Uppsala, Sweden, 8 School of Sports Science, UiT The Arctic University of Norway, Tromsø, Norway, 9 Department of Physical Education and Sports, Faculty of Sport Sciences, Sport and Health University Research Institute (iMUDS), University of Granada; CIBEROBN, ISCIII, Granada, Andalucía, Spain, 10 Faculty of Sport and Health Sciences, University of Jyväskylä, Jyväskylä, Finland, 11 Wellbeing Services County of Central Finland, Jyväskylä, Finland, 12 Department of Biomedicine, Aarhus University, Aarhus, Denmark, 13 Institute of Environmental Medicine, Karolinska Institutet, Stockholm, Sweden

‡ These authors are joint senior authors on this work.
* marcel.ballin@uu.se

**Academic editor:** Andre P. Kengne, South African Medical Research Council, SOUTH AFRICA

## Abstract

### Background

Cardiorespiratory fitness has been linked to both lower and higher risks of cancer, but the evidence comes from observational analysis which may be influenced by unobserved confounders and bias processes. We aimed to examine the associations between adolescent cardiorespiratory fitness and risk of cancer in late adulthood while addressing the unknown influence of unobserved familial confounders and diagnostic bias processes.

### Methods and findings

We conducted a sibling-controlled cohort study with registry linkage based on all Swedish men who participated in mandatory military conscription examinations from 1972 to 1995 and who completed standardized cardiorespiratory fitness testing. The outcomes were overall cancer diagnosis and cancer mortality, and 14 site-specific cancers (diagnosis or death), ascertained using the National Patient Register and Cause of Death Register until 31 December 2023. A total of 1,124,049 men, including 477,453 full siblings, with a mean age of 18.3 years at baseline, were followed until a median (maximum) age of 55.9 (73.5) years, during which 98,410 were diagnosed

**Data availability statement:** The data in this study are not available to the public and will not be shared according to regulations under Swedish law. Researchers interested in obtaining the data may seek ethical approvals and inquire through Statistics Sweden. For further advice see: https://www.scb.se/en/services/ordering-data-and-statistics/. The analytical code is available at GitHub (https://doi.org/10.5281/zenodo.14673830).

**Funding:** MN is funded via grants from the Swedish Research Council (2019-00738). VHA is funded via grants from the National Institute for Aging and the National Institute of Neurological Disorders and Stroke (1R01NS131433-01). The funders had no role in study design, data collection and analysis, decision to publish, or preparation of the manuscript.

**Competing interests:** I have read the journal's policy and the authors of this manuscript have the following competing interest: MB is employed at the Swedish Medical Products Agency, SE-751 03 Uppsala, Sweden. The views expressed in this paper do not necessarily represent the views of this Government agency. MN reported serving on advisory boards for Johnson & Johnson and Itrim, and serving as a consultant for the Armed forces. No other disclosures were reported.

**Abbreviations:** BMI, body mass index; CI, confidence interval; HRs, hazard ratios; PAFs, population-attributable fractions; RECORD, Reporting of Studies Conducted using Observational Routinely-Collected Data.

with cancer and 16,789 died from cancer (41,293 and 6,908 among full siblings respectively). In cohort analysis, individuals in the highest quartile of fitness had a lower risk of overall cancer mortality (adjusted hazard ratio [HR]: 0.71, 95% confidence interval [CI] 0.67, 0.76; $P<0.001$) compared to the lowest quartile, corresponding to a standardized cumulative incidence (1-Survival) difference of −0.85 (95% CI [−1.00, −0.71]) percentage points at 65 years of age. Individuals in the highest quartile of fitness also had lower risks (HRs ranging from 0.81 to 0.49, incidence differences ranging from −0.13 to −0.32 percentage points; $P<0.001$ for all) of rectum, head and neck, bladder, stomach, pancreas, colon, kidney, liver, bile ducts, and gallbladder, esophagus, and lung cancer. Yet, individuals in the highest quartile of fitness had higher risks of prostate (HR: 1.10, 95% CI [1.05, 1.16]; $P<0.001$, incidence difference: 0.48 percentage points, 95% CI [0.23, 0.73]) and skin cancer (e.g., non-melanoma HR: 1.44, 95% CI [1.38, 1.50]; $P<0.001$, incidence difference: 1.84 percentage points, 95% CI [1.62, 2.05]). Individuals in the highest quartile of fitness had a higher risk of overall cancer diagnosis (HR: 1.08, 95% CI [1.06, 1.11]; $P<0.001$, incidence difference: 1.32 percentage points, 95% CI [0.94, 1.70]), results driven by prostate and skin cancer being the most common types of cancer. When comparing full siblings, and thereby controlling for unobserved shared confounders, the lower risk of overall cancer mortality remained (HR: 0.78, 95% CI [0.68, 0.89]; $P<0.001$, incidence difference: −0.61 percentage points, 95% CI [−0.93, −0.28]), while the excess risk of prostate (HR: 1.01, 95% CI [0.90, 1.13]; $P=0867$, incidence difference: 0.05 percentage points, 95% CI [−0.50, 0.60]), skin (e.g., non-melanoma HR: 1.09, 95% CI [0.99, 1.20]; $P=0.097$, incidence difference: 0.40 percentage points, 95% CI [−0.07, 0.87]), and overall cancer diagnosis (HR: 1.00, 95% CI [0.95, 1.06]; $P=0.921$, incidence difference: 0.04 percentage points, 95% CI [−0.80, 0.88]) attenuated to the null. For other site-specific cancers, sibling comparisons results varied, with more attenuation for melanoma, kidney, stomach, bladder, pancreas, and liver, bile ducts, and gallbladder cancer, while associations with lung, colon, head and neck, and esophagus cancer seemed to attenuate less. The findings were confirmed through an extensive set of sensitivity analyses. The main limitations of this study include the lack of inclusion of female participants, lack of data on other risk factors such as smoking, alcohol consumption, and physical activity, and only adjustment for the unobserved confounders which are shared between full siblings.

## Conclusions

Higher levels of adolescent cardiorespiratory fitness were associated with lower overall cancer mortality in late adulthood, a finding that persisted in sibling comparisons. However, the influence of unobserved familial confounding appeared to vary by cancer type and be more pronounced for cancer diagnoses than for mortality. This may suggest a need for robust causal methods to triangulate results, rather than relying on correlations alone, to better inform public health efforts.

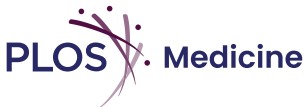

## Author summary

### Why was this study done?

- High cardiorespiratory fitness, a measure of regular physical activity and linked to genetics, has been associated with reduced cancer risk.

- Improving fitness in young people is a growing public health focus.

- Existing studies may have limitations, such as not fully accounting for other factors, such as participation in cancer screening and other health behaviors that may be more common in individuals with higher fitness levels.

### What did the researchers do and find?

- We studied over 1.1 million young men, including nearly 500,000 full brothers, using data from fitness tests at military conscription (around age 18) linked to Swedish healthcare and death registries to track cancer diagnoses and deaths in adulthood.

- We found that high cardiorespiratory fitness in adolescence was associated with lower overall cancer-related deaths. This link remained even after accounting for factors shared between siblings, such as genetics, shared environment, and health-seeking behaviors such as participation in cancer screening.

- We also found varying confounding by specific cancer sites, where associations between cardiorespiratory fitness and increased diagnoses of prostate and skin cancer appeared to be driven by other factors, as they were not observed when comparing full siblings. Associations between high cardiorespiratory fitness and lower risks of kidney, stomach, bladder, pancreas, and liver, bile ducts, and gallbladder cancer also became smaller after accounting for factors shared between siblings.

### What do these findings mean?

- If supported by further causal analyses, our study suggests that while high adolescent cardiorespiratory fitness might protect against certain cancers, other associations might be driven by unmeasured factors or biases.

- Our findings could highlight the importance of well-designed research when studying cardiorespiratory fitness and cancer risk for an accurate appreciation of potential public health benefits.

- Our study did not include female participants, was not able to account for certain relevant risk factors for cancer such as smoking, alcohol, and physical activity, and only accounted for the unobserved confounders that are shared between full siblings. These limitations should be addressed in future studies to expand upon our results.

## Introduction

The International Agency for Research in Cancer states that physical activity is a proven way to lower the risk of cancers [1], a statement also reflected through the World Health Organization guidelines for physical activity [2]. Recently, cardio-respiratory fitness, a marker influenced by long-term physical activity and genetic factors [3,4], has also been implicated in cancer risk [5–9]. This has been partly attributed to its moderating effect on the relationship between adiposity and cancer mortality [5], its influence on endogenous sex hormones, metabolic factors, growth factors, immune-related pathways [10], inflammation reduction [11], as well as its potential to reduce the risk of other somatic diseases that may impact cancer prognosis [10]. Considering the challenges in identifying novel modifiable risk factors for cancer prevention [12], along

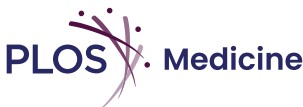

with the downward trends in physical activity and cardiorespiratory fitness levels especially among young people [13–16], public health interventions targeting these traits in adolescents are gaining traction as a means of cancer prevention.

Unfortunately, the current evidence supporting the role of physical activity and cardiorespiratory fitness in cancer is associated with low certainty [9,17]. One major limitation is that existing large-scale studies have been unable to capture important differences in behavioral, environmental, and genetic confounders. This limitation is compounded by the fact that existing studies often used a composite endpoint of diagnoses and deaths, which could introduce detection bias given that individuals with high levels of physical activity more often have different health-seeking behaviors [18], such as higher participation in cancer screening programs [19,20]. This increases the likelihood of early diagnosis, including for cancers typically associated with low mortality (e.g., non-melanoma skin cancer). When this is bundled together with cancer mortality, a protective effect on cancers associated with higher mortality may be underestimated or lost in the mix. Some studies have even found that higher physical activity and cardiorespiratory fitness are associated with a higher risk of overall cancer [6,21–24], and especially skin and prostate cancer [6,7,20,25], probably reflecting such bias.

These differential health behaviors remain difficult to capture and address in large-scale observational studies. One possible way to account for unobserved factors is to compare family members, such as siblings [26]; who are assumed to share some of these behaviors and other important genetic and environmental confounders [27,28]. Of prior studies on the topic of fitness and cancer, including several studies using Swedish registry data [6,7,20,27,29–31], including the conscription, there has, to our knowledge, only been one which attempted to account for unobserved confounders [27]. However, this study by Högström and colleagues only reported time-specific hazard ratios (HRs) [27], which are difficult to interpret as they are largely influenced by the depletion of susceptibles [32], and site-specific cancers were not examined. As such, additional family-based studies are clearly warranted.

In this study, we aim to address the existing inconsistencies and establish a more accurate appreciation for the cancer prevention potential of cardiorespiratory fitness. We conducted a nationwide cohort study over six decades, encompassing over 1.1 million men, to examine the associations between adolescent cardiorespiratory fitness and risk of cancer in late adulthood. To address the influence of unobserved familial confounders and bias processes, we performed sibling-comparison analyses in nearly 0.5 million full brothers.

## Methods

### Study design

A cohort study with full-sibling analysis was conducted by cross-linking data from Swedish health and population registers, using the Personal Identification Number, which is unique for all Swedish residents [33]. The need for informed consent was waived by the Swedish Ethical Review Authority as the study only leveraged data from existing registers [34]. The study was approved by the Regional Ethical Review Board in Umeå (no. 2010-113-31M). This study is reported as per the Reporting of Studies Conducted using Observational Routinely-Collected Data (RECORD) guideline (S1 Checklist: RECORD Checklist).

### Databases

From the Swedish Military Service Conscription Register [35], we obtained an eligible study population based on all men who participated in military conscription examinations between 1972 and 1995, during which cardiorespiratory fitness was assessed. During this period, conscription around the age of 18 years was mandatory for all males in Sweden, with few exemptions (approximately 90% population coverage) [35]. To this, family relationships were identified using the Multi-Generation Register [36], subsequent mortality using the National Cause of Death Register [37], inpatient and specialist outpatient care using the National Patient Register [38], and emigration and socioeconomic data using the registries of Statistics Sweden [36]. Overall, these data cover the entire Swedish population and are mandated by law.

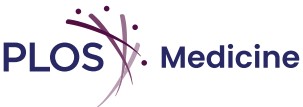

### Derivation of study population

A total of 1,249,131 men were conscripted between 1972 and 1995. From these, we excluded 33,645 (2.7%) with missing data on cardiorespiratory fitness, 72,042 (5.8%) with missing data on the covariates, and 19,395 (1.5%) with extreme values (detailed below). This resulted in 1,124,049 conscripts being included in the cohort analysis (90.0% retained). Among these, 477,453 were full siblings from 219,304 families and were included in the sibling analysis. S1 Fig shows the participant flow, including the distribution of missing data.

### Exposure

Cardiorespiratory fitness was assessed at conscription using validated equipment and a standardized procedure as described in detail elsewhere [35]. In short, conscripts performed a maximal ergometer bicycle test following normal electrocardiography. The conscript began cycling for 5 min at 60–70 revolutions per minute at a low level of predetermined (by body weight) resistance. The external resistance was then gradually increased by 25 watts every minute until exhaustion. The test results were recorded as the maximum number of watts of resistance achieved ($W_{max}$), which correlates strongly with maximal oxygen uptake [39]. Similar to previous studies, we excluded conscripts with extremely low values (<100 $W_{max}$) [28,40].

### Outcomes

The primary outcomes were overall cancer diagnosis and cancer mortality, as recorded in the National Patient Register and the Cause of Death register until 31 December 2023. To this, we also perform an analysis of 14 site-specific cancers (either diagnosis or death), which were selected based on their previously identified associations with physical activity or cardiorespiratory fitness:[2,5,6,21] head and neck, esophagus, lung, stomach, pancreas, colon, rectum, kidney, prostate, bladder, myeloma, skin (divided into melanoma and non-melanoma), and liver, bile ducts, and gallbladder (enhancing comparability with a previous study [6]). A diagnosis was ascertained as the date of first hospitalization or visit in specialized outpatient care, recorded using diagnostic codes in a primary or secondary position, and death was classified as cancer-related if it was recorded as the primary or secondary cause on the death certificate (S1 Table).

### Covariates

From the Swedish Military Service Conscription Register, we collected data on age at conscription, year of conscription, and measured body mass index (BMI, kg/m$^2$). As in previous studies, conscripts with extreme BMI values (≤15 and ≥60 kg/m$^2$) were excluded [28,41]. From Statistics Sweden [42], we obtained data on the socioeconomic status of both mothers and fathers of the conscripts, including information on the highest attained lifetime education and the annual disposable income standardized by birth years into highest-achieved quintiles between ages 40 and 50 (used to capture working life income, henceforth referred to as income categories). When values for both the mother and father were available, the highest value was retained.

### Statistical analysis

There was no prospectively written analysis plan for the present study. Participants were followed from the day of conscription until the date of a cancer outcome, emigration, death from non-cancer causes, or end of follow-up (31 December 2023), whichever came first. The sibling analysis was based on the same start of follow-up and end points as the cohort analysis. Flexible parametric survival models were used to estimate HRs with 95% confidence intervals (CIs) for each outcome by levels of cardiorespiratory fitness, using age as the underlying time scale, and baseline knots placed at the 5th, 27.5th, 50th, 72.5th, and 95th percentile of the uncensored log survival times [43,44]. We based the knot placement on Harrell's recommendations [44]. These models yield similar HRs as the conventional Cox regression, while having the

additional advantage of directly estimating the baseline hazard, thereby enabling the computation of absolute effects measures [43]. We modeled cardiorespiratory fitness both as quartiles and using restricted cubic splines with knots placed at the 5th, 35th, 65th, and 95th percentile [44]. Since HRs are difficult to clinically contextualize [32], we also calculated the standardized cumulative incidence at 65 years of age (1-Survival) and illustrated it graphically over the follow-up period. We adjusted for observed covariates which we assumed, based on existing literature and clinical expertise, to be causally related to both the exposure and the outcome (i.e., potential confounders) (S2 Fig), including age at conscription (continuous), year of conscription (1972, 1973–1977, 1978–1982, 1983–1987, 1988–1992, and 1993–1995), BMI (continuous and quadratic term), parental education (compulsory school ≤9 years, secondary education, post-secondary education <3 years, post-secondary education ≥3 years), and parental income (five categories).

For the full-sibling analysis, the aforementioned model was extended to a marginalized between-within model with robust standard errors, which enables control for all unobserved confounders (including genetic and shared environmental factors) shared between the siblings [26,45]. The between-within model isolates the individual-level variation (within effect) from the family-level variation (between effect) by including a term for the exposure/covariate and a term for its family average [26,45]. We have previously employed these methods for time-to-event analysis [28].

To further aid the interpretation of potential changes in estimates from cohort analysis to sibling analysis, we estimated population-attributable fractions (PAFs) [46], considering a "major" intervention that would shift everyone to the top quartile of cardiorespiratory fitness, or a "moderate" intervention that would shift everyone who belonged to the bottom quartile to the second quartile. All analyses were performed using Stata MP version 18.

**Sensitivity analyses.** A series of sensitivity analyses were conducted to test the robustness of the results. First, to examine whether differences in estimates in sibling analysis as compared to cohort analysis were more likely to be due to selection bias rather than control for unobserved shared confounders, the standard analysis was replicated in the sibling cohort (i.e., without controlling for shared confounders) [26]. Second, because BMI could be considered both a confounder or mediator in the associations between cardiorespiratory fitness and cancer [47], the main analysis was repeated but excluding control for BMI. Third, to examine the influence of death from non-cancer causes as a competing event, the aforementioned flexible parametric models were used to compute cause-specific cumulative incidence functions (1-Survival) of all cancer outcomes at 65 years of age, accounting for death from non-cancer causes [48].

The following additional sensitivity analyses were conducted after suggestions from peer reviewers. First, we repeated our analysis using Cox regression, to facilitate a comparison between the modeling approaches. Second, in addition to adjusting for BMI and excluding it due to potential mediation, we conducted an analysis incorporating World Health Organization BMI categories (i.e., underweight <18.5 kg/m², normal weight 18.5–24.9 kg/m², overweight 25.0–29.9 kg/m², obesity ≥30.0 kg/m²), and their interaction terms with cardiorespiratory fitness quartiles, because BMI may effect-modify the relationship between cardiorespiratory fitness and the outcomes. We computed marginal HRs across BMI categories, allowing for interaction, and post-estimated HRs within each BMI category. Third, in the primary analysis, we excluded conscripts with missing cardiorespiratory fitness and covariates (complete-case analysis), as we suspected the data be missing completely at random or not at random. As an alternative, we applied multiple imputation with chained equations ($K = 20$ repetitions) to retain these individuals. The procedure was performed separately for the total cohort and full siblings, using linear and multinomial logistic models for continuous (BMI) and categorical parameters (quartiles of cardiorespiratory fitness, parental education, and income). Age at conscription, conscription year, follow-up time, and censoring were used as complete auxiliaries, while weight and length at conscription were treated as partially observed auxiliaries (imputed using a linear model). Fourth, we also repeated our primary analysis (based on 5 knots) varying the number of knots for the baseline hazard between 3 and 7, basing the placement on Harrell's recommendations [44]. Fifth, we computed time-specific HRs for every 5-year interval from age 30 to 65 for all outcomes, allowing the HRs to vary as a function of the follow-up time. Finally, despite few cases (S4 Table), we also computed HRs for site-specific mortality in cohort

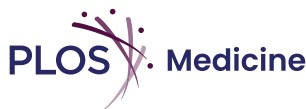

analysis for 12 of the 14 sites where we observed >300 censoring events (excluding non-melanoma skin and bladder), as these may still be valuable for future meta-analyses.

## Results

### Baseline characteristics

Baseline characteristics were similar in the total cohort and in the sibling cohort (Table 1). The mean (standard deviation) age at conscription was 18.3 (0.7) years and most (81.3%) were normal weight. About a quarter had parents with a high (post-secondary) level of education and about a third had parents with a high level of annual income. Overall, baseline characteristics were similar in the analytical sample as compared to the total dataset of all available conscripts with no restrictions and after stepwise exclusions for year of conscription, missing fitness data, missing covariate data, and extreme values (S2 Table).

### Cancers during follow-up

During a median (range) follow-up of 37.7 (0.1, 51.4) years, 98,410 (8.8%) were diagnosed with cancer and 16,789 (1.5%) died of cancer in the total cohort, and 41,293 (8.7%) and 6,908 (1.5%) in the sibling cohort. The median (range) age at first cancer diagnosis was 55.1 (19.1, 73.0) years in the total cohort and 54.8 (19.1, 72.6) years in the sibling cohort. The median (range) age at cancer death was 55.7 (18.4, 72.9) years in the total cohort and 55.3 (18.4, 72.9) years in the sibling cohort. The most common site-specific cancers were non-melanoma skin (27,105 diagnoses and 227 deaths) and prostate cancer (24,211 diagnoses and 869 deaths) (Tables 1 and S3). The number of cancer-specific deaths are described in S4 Table.

### Cardiorespiratory fitness and overall cancer

In cohort analysis, high cardiorespiratory fitness was associated with a higher risk of cancer diagnosis (Figs 1 and S3 and S5 and S6 Tables). Compared to the bottom quartile of cardiorespiratory fitness, the HR in the top quartile was 1.08 (95% CI [1.06, 1.11]; $P<0.001$). The standardized cumulative incidence at 65 years of age was 18.04% (95% CI [17.78, 18.30]) in the bottom quartile and 19.36% (95% CI [19.06, 19.67]) in the top quartile, with a difference of 1.32 (95% CI [0.94, 1.70]) percentage points. Calculations of PAFs produced negative estimates, implying a theoretical increase in the proportion of cancer diagnoses should either a moderate or major intervention be carried out (S7 Table). In full-sibling analysis, the association attenuated to the null (HR: 1.00, 95% CI [0.95, 1.06]; $P=0.921$, incidence difference: 0.04 percentage points (95% CI [−0.80, 0.88]).

In cohort analysis, high cardiorespiratory fitness was associated with a lower risk of cancer mortality (Figs 1 and S3 and S5 and S6 Tables). Compared to the bottom quartile, the HR in the top quartile was 0.71 (95% CI [0.67, 0.76]; $P<0.001$). The standardized cumulative incidence at 65 years of age was 3.01% (95% CI [2.91, 3.11]) in the bottom quartile and 2.15% (95% CI [2.05, 2.25]) in the top quartile, with a difference of −0.85 (95% CI [−1.00, −0.71]) percentage points. The PAF of cancer mortality associated with a moderate intervention was 21.4% (95% CI [18.8, 24.1]) and for a major intervention 28.4% (95% CI [24.3, 32.6]) (S7 Table). In full-sibling analysis, the association modestly attenuated (HR: 0.78, 95% CI [0.68, 0.89], incidence difference: −0.61 percentage points, 95% CI [−0.93, −0.28]). As such, the PAF associated with a moderate intervention dropped to 14.8% (95% CI [8.0, 21.6]), and for a major intervention 21.6% (95% CI [11.3, 32.0]).

### Cardiorespiratory fitness and site-specific cancer

In cohort analysis, high cardiorespiratory fitness was associated with a lower risk (HRs ranging from 0.81 to 0.49) of the rectum, head and neck, bladder, stomach, pancreas, colon, kidney, liver, bile ducts, and gallbladder, esophagus, and lung

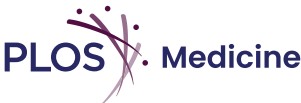

**Table 1.** Baseline characteristics and number of cancer outcomes during follow-up in the total cohort and the full sibling cohort.

| | Total cohort (*N*=1 124,049) | Full sibling cohort (*N*=477,453) |
|---|---|---|
| **Baseline data** | | |
| **Birth year, median (IQR)** | 1966 (1960–1971) | 1965 (1961–1970) |
| **Age at conscription, mean (SD)** | 18.3 (0.7) | 18.3 (0.7) |
| **Body mass index, kg/m²** | | |
| Median (range) | 21.4 (15.0–59.2) | 21.3 (15.0–59.2) |
| **Body mass index categories, *n* (%)** | | |
| Underweight (<18.5 kg/m²) | 88,966 (7.9) | 37,884 (7.9) |
| Normal weight (18.5–24.9 kg/m²) | 913,586 (81.3) | 390,811 (81.9) |
| Overweight (25.0–29.9 kg/m²) | 102,267 (9.1) | 41,248 (8.6) |
| Obesity (>30.0 kg/m²) | 19,230 (1.7) | 7,510 (1.6) |
| **Cardiorespiratory fitness, $W_{max}$[a]** | | |
| Median (range) | 271 (100–999) | 271 (100–999) |
| **$W_{max}$[a] by quartiles, median (range)** | | |
| Quartile 1 | 217 (100–236) | 217 (100–236) |
| Quartile 2 | 253 (237–270) | 253 (237–270) |
| Quartile 3 | 290 (271–312) | 290 (271–312) |
| Quartile 4 | 339 (313–999) | 339 (313–999) |
| **Parental level of education, *n* (%)** | | |
| Compulsory school <9 years | 319,242 (28.4) | 139,246 (29.1) |
| Secondary education | 505,510 (45.0) | 210,246 (44.0) |
| Post-secondary education <3 years | 123,757 (11.0) | 50,210 (10.5) |
| Post-secondary education >3 years | 175,540 (15.6) | 77,751 (16.3) |
| **Parental highest income, *n* (%)** | | |
| Category 1 (low income) | 55,055 (4.9) | 18,918 (4.0) |
| Category 2 | 108,836 (9.7) | 41,820 (8.8) |
| Category 3 | 237,953 (21.2) | 101,150 (21.2) |
| Category 4 | 338,444 (30.1) | 145,735 (30.5) |
| Category 5 (high income) | 383,761 (34.1) | 169,830 (35.6) |
| **Number of cancer events (median age at event) during follow-up** | | |
| Overall cancer diagnosis | 98,410 (55.1) | 41,293 (54.8) |
| Overall cancer mortality | 16,789 (55.7) | 6,908 (55.3) |
| Head and neck | 4,026 (54.1) | 1,692 (53.7) |
| Esophagus | 1,178 (57.1) | 464 (57.0) |
| Lung | 3,131 (57.4) | 1,263 (57.1) |
| Stomach | 1,430 (55.4) | 599 (55.3) |
| Pancreas | 2,255 (57.2) | 943 (57.0) |
| Liver, bile ducts, and gallbladder | 2,246 (57.1) | 925 (56.8) |
| Colon | 5,320 (55.1) | 2,177 (54.8) |
| Rectum | 3,917 (55.7) | 1,625 (55.6) |
| Kidney | 2,741 (54.6) | 1,151 (54.5) |
| Prostate | 24,225 (59.5) | 10,042 (59.1) |
| Bladder | 3,490 (56.9) | 1,432 (56.3) |
| Myeloma | 1,460 (55.8) | 580 (55.9) |

*(Continued)*

**Table 1.** (Continued)

| | Total cohort (*N*=1 124,049) | Full sibling cohort (*N*=477,453) |
|---|---|---|
| Melanoma skin | 10,026 (52.5) | 4,216 (52.5) |
| Non-melanoma skin | 27,302 (54.5) | 11,509 (54.2) |

ᵃ $W_{max}$ is the maximum number of watts of resistance achieved on a maximal ergometer bicycle test with gradually increasing resistance.

IQR, interquartile range; SD, standard deviation; $W_{max}$, watt maximum.

cancer (Figs 2, S3–S5 and S5 and S6 Tables). In contrast, high cardiorespiratory fitness was associated with a higher risk of prostate (HR: 1.10, 95% CI [1.05, 1.16]; *P*<0.001, incidence difference: 0.48 percentage points, 95% CI [0.23, 0.73]), melanoma (HR: 1.50, 95% CI [1.41, 1.61]; *P*<0.001, incidence difference: 0.80 percentage points, 95% CI [0.67, 0.94]), and non-melanoma skin cancer (HR: 1.44, 95% CI [1.38, 1.50]; *P*<0.001, incidence difference: 1.84 percentage points, 95% CI [1.62, 2.05]).

In full-sibling analysis, the degree of change in the associations, as compared to in the total cohort, varied between cancer types (Figs 2, S3–S5 and S5 and S6 Tables). The excess risk of non-melanoma skin cancer was largely attenuated (HR: 1.09, 95% CI [0.99, 1.20]; *P*=0.097, incidence difference: 0.40 percentage points, 95% CI [−0.07, 0.87]), as was the risk of prostate cancer (HR: 1.01, 95% CI [0.90, 1.13]; *P*=0.867, incidence difference: 0.05 percentage points, 95% CI [−0.50, 0.60]). Attenuation was observed also for melanoma skin cancer, liver, bile ducts, and gallbladder cancer, kidney cancer, stomach cancer, bladder cancer, and pancreas cancer (S5 and S6 Tables), translating into reduced PAFs (S7 Table). For lung cancer, there was no attenuation, and for colon, head and neck, and esophagus cancer, the HRs were also similar, albeit associated with larger uncertainty and not necessarily statistically significant (S5 and S6 Tables). The estimates for myeloma were not statistically significant either in cohort or sibling analysis (S5 and S6 Tables).

## Sensitivity analyses

Replication of the standard analysis in the sibling cohort yielded overall similar estimates as observed in the cohort analysis (S8 Table). The exception was rectum cancer, for which the estimates were more similar to those observed in sibling analysis. Excluding adjustment for BMI resulted in similar or slightly weaker associations for most outcomes, with a similar pattern in the cohort and sibling analysis (S9 Table). Accounting for the competing risk of non-cancer death (so-called crude risks) was similar to standard analysis across most outcomes (S10 Table). The exceptions were overall cancer diagnosis and site-specific cancers of the prostate and skin, for which the crude risks were slightly smaller. The HRs were consistent using Cox regression instead of flexible parametric models (S11 Table). The findings also remained largely consistent when accounting for interactions between BMI and cardiorespiratory fitness (S12 Table). However, BMI-strata-specific estimates were limited by low statistical power, as most conscripts were normal weight. Using multiple imputations with chained equations to impute individuals excluded due to missing covariate data yielded results consistent with the main analysis based on complete cases (S13 Table). The findings were consistent when using a different number of knots for the baseline hazard (S14 and S15 Tables). When the associations were estimated in 5-year intervals, higher fitness was associated with a higher risk of certain cancers at a young age (e.g., age 30), which generally attenuated toward the null in the sibling analysis (S16 Table). Finally, albeit estimated with greater uncertainty, analysis of cancer-specific mortality in the total cohort yielded similar findings as when using a composite endpoint of diagnosis and deaths for many cancers (S17 Table). The exception was melanoma skin and prostate cancer, for which the excess risks observed in the composite analysis were closer to unity for the former (e.g., top versus bottom quartile HR: 1.06, 95% CI [0.84, 1.34]; *P*=0.633), and protective for the latter (e.g., top versus bottom quartile HR: 0.87, 95% CI [0.65, 1.16]; *P*=0.335) (S17 Table).

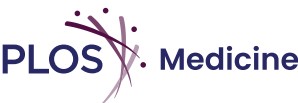

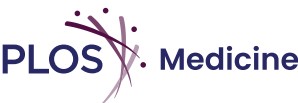

**Fig 1. Standardized cumulative incidences and hazard ratios for overall cancer diagnosis and cancer mortality by quartiles of cardiorespiratory fitness in cohort and sibling analysis.** CI, confidence interval. Estimates obtained using flexible parametric survival models, extended to a

marginalized between-within model in the sibling cohort, with baseline knots placed at the 5th, 27.5th, 50th, 72.5th, and 95th percentile of the uncensored log survival times, and using age as the underlying time scale. The bottom quartile was the referent. All models were adjusted for age at conscription, year of conscription, body mass index, parental education, and parental income. Inferential measures for the incidences were omitted for clarity as they are also reflected by the hazard ratios and are reported in S5 and S6 Tables.

## Discussion

In this nationwide cohort study, we found that higher levels of adolescent cardiorespiratory fitness were associated with lower overall cancer mortality in late adulthood, and this association replicated when comparing full siblings. At the same time, we found evidence suggesting that the associations between adolescent fitness and later-life cancer risk may be affected by familial confounding factors, and there appeared to be varying confounding by specific cancer sites.

Physical activity and cardiorespiratory fitness have been linked to lower cancer risk, although the certainty in the evidence is limited, including a largely unknown contribution of unobserved confounding and bias processes [5–9,21,27]. In this study, high adolescent cardiorespiratory fitness was linked to excess risk of non-melanoma skin and prostate cancer, as in previous observational studies and meta-analyses [6,21]. However, we provide evidence to suggest that this risk may be explained by bias as it could not be replicated after accounting for unobserved confounders shared between full siblings. This is further supported by the analysis of prostate-related mortality, where, despite lower statistical power, cardiorespiratory fitness was not associated with an excess risk (e.g., top versus bottom quartile HR: 0.87, 95% CI [0.65, 1.16]). The sibling analysis cannot pinpoint the exact confounding factor(s), but it may be a combination of differential health-seeking behaviors (including participation in cancer screening), as well as other shared environmental factors and genetics. Previous studies showing associations between physical activity and cardiorespiratory fitness and excess risk of skin and prostate cancer have often suggested that their findings may be the result of bias [6,20,49], speculating that it may be explained by fit people also spending more time outdoors in the sun and that they participate in screening programs to a greater extent [50,51]. Evidence rendering support to these confounding and bias processes has, however, been lacking until now. Yet, even in this large-scale study, the excess risk of melanoma skin cancer could not be fully explained by confounders shared between full siblings. This may suggest the presence of biases beyond those shared by siblings, and we encourage future research to further scrutinize this association.

The association between high adolescent cardiorespiratory fitness and lower risk of overall cancer mortality remained statistically significant after accounting for such confounding factors (HR: 0.78, 95% CI [0.68, 0.89]). In a previous study based on the same population, Högström and colleagues reported a lower risk of both cancer diagnosis and cancer mortality in sibling analysis [27]. Yet, interpretation of those estimates and comparisons to ours is difficult considering that the study relied on a younger cohort with substantially fewer cancer diagnoses and deaths (15,093 [1.3%] and 4,900 [0.4%], compared to our study with 98,410 [8.8%] diagnoses and 16,789 [1.5%] deaths). [27] Additionally, only time-specific HRs were reported [27], which are difficult to interpret as they are largely influenced by the depletion of susceptibles [32]. Nevertheless, based on the results presented here, it appears important to distinguish a diagnosis of cancer from cancer-related mortality since collapsing the two may dilute the benefits associated with high-mortality cancers by introducing biased associations with a diagnosis of low-mortality cancers detected in screening. By extension, this could suggest that previous large-scale studies and meta-analyses of cardiorespiratory fitness or physical activity, which relied on composite endpoints, might have underestimated the protective associations [5,6,21,23,24]. If this is true, and can be verified by future causal analysis, the potential of public health initiatives targeting cardiorespiratory fitness or physical activity to reduce cancer mortality might be underappreciated [52,53].

This study also revealed varying confounding by specific cancer sites. For example, the associations showing lower risks of kidney, stomach, bladder, pancreas, and liver, bile ducts, and gallbladder cancer seemed to attenuate more. In contrast, associations with colon, head and neck, and esophagus cancer seemed to attenuate less, although these

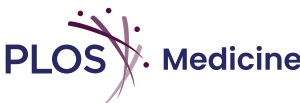

**Fig 2. Hazard ratios for diagnosis or death from site-specific cancers by quartiles of cardiorespiratory fitness in cohort and sibling analysis.**
CI, confidence interval. Estimates were obtained using flexible parametric survival models, extended to a marginalized between-within model in the

sibling cohort, with baseline knots placed at the 5th, 27.5th, 50th, 72.5th, and 95th percentile of the uncensored log survival times, and using age as the underlying time scale. The lowest quartile was the referent. All models were adjusted for age at conscription, year of conscription, body mass index, parental education, and parental income.

associations were characterized by larger uncertainty. These findings may suggest a more nuanced role of adolescent cardiorespiratory fitness in site-specific cancer, whereby there may be a greater preventive potential for certain types and less for others. Assuming part of the attenuation is the result of control for familial confounders, these findings align with research showing a clear variation in the contribution of genetic and shared environmental factors to the risk of different cancers [54,55]. Of note is that high adolescent cardiorespiratory fitness was strongly associated with a reduced risk of lung cancer, even when comparing full siblings. The World Health Organization has claimed there to be insufficient evidence to determine whether the association between high physical activity and reduced risk of lung cancer is causal or non-causal due to confounding by smoking [2]. Although we were unable to adjust directly for smoking, our sibling analysis likely partly captured smoking behavior and secondary smoking exposure through control for shared environmental factors, given their large influence on smoking [56]. However, had smoking and other behaviors (e.g., alcohol intake) been directly measured at conscription we could have empirically examined their confounding potential on the relationship between cardiorespiratory fitness and cancer. Nevertheless, these findings also extend those of a previous meta-analysis of four small studies wherein smoking was adjusted for [5].

This study has several notable strengths, including its nationwide coverage, family-based analysis, standardized measures of cardiorespiratory fitness, extensive follow-up until late adulthood with virtually zero attrition, and investigation of both diagnoses of cancer in a universal healthcare system and cancer mortality.

However, some limitations should be considered. First, our study was limited to men due to historical conscription practices in Sweden. While some associations could be hypothesized to be similar, we emphasize the importance of replicating our findings in female populations, particularly given sex-specific cancer rates and cancers unique to females. Such replication efforts should prioritize using robust causal methods to address the apparent important role of unobserved confounding. Second, our study focuses on adolescent cardiorespiratory fitness and may provide insights into public health policies aimed at young individuals. However, it may not directly inform interventions targeted at other populations, despite evidence that cardiorespiratory fitness moderately tracks from adolescence into adulthood [57]. Third, due to the limited number of mortality censoring events for site-specific cancers, we were unable to estimate cancer-specific mortality with high confidence; longer follow-up will be necessary to improve the precision of these estimates, and to enable such analyses using sibling comparisons. Furthermore, the small counts of some site-specific cancers can explain the CIs overlapping the null in the sibling analysis, which is not necessarily evidence of the absence of effects. Fourth, we were unable to adjust our analyses for certain important risk factors such as smoking and alcohol consumption, which were not available in the databases used. Additionally, a limitation is the lack of data on physical activity, which would have been valuable to explore its potential as a confounder in the association between fitness and cancer. While there have been many suggested mechanisms thought to explain the link between fitness and cancer risk [5,10,11], it is possible that physical activity also has independent links to cancer risk. As outlined in a recent review, however, there is a need for more mechanistic studies on this topic [11]. We welcome future studies to attempt to distinguish between the role of activity versus fitness in cancer risk. Fifth, while comparisons of full siblings are assumed to control for several important confounders that are difficult to measure at scale, this design hinges on assumptions [26,58] (e.g., absence of carry-over effects, measurement error, and non-shared confounding) and there likely remains residual confounding (e.g., the remaining 50% of genetics not shared between full siblings and behaviors not shared by full siblings with different fitness levels). Future studies utilizing monozygotic twins or other causal designs would be beneficial for triangulating these findings.

In our study, we examined adolescent cardiorespiratory fitness, but another question is how variations in fitness across the life span influence cancer risk [59]. To our knowledge, no existing studies have explored the role of unobserved confounding in such questions, which we believe would be a valuable area for future research. An interesting and related question is whether factors present even before adolescence could be incorporated to provide a deeper understanding of the life course trajectory of risk factors. Such studies may become feasible as the youngest individuals in the current cohort—who have data on factors at birth [60,61] —reach ages where the incidence of somatic diseases is expected to rise. Additionally, while our study provides insight into adolescent cardiorespiratory fitness, it would be valuable to apply similar causal approaches to estimate the influence of measured adolescent physical activity on future cancer risk. To our knowledge, existing studies have primarily relied on standard observational analysis [21], with no examination of unobserved confounding in relation to physical activity.

The implications of these findings may be considered in the light of the downward trends in population levels of cardiorespiratory fitness, which is observed across all ages, and especially in young people [13–15]. Our findings alone are not sufficient to inform definitive policy statements as they need to be corroborated by further causal analyses. Yet, our findings suggest that the influence of behavioral, environmental, and genetic confounders shared between full siblings might only partly account for the beneficial association between higher fitness and lower cancer mortality. For instance, the hypothetically preventable share of cancer mortality at 65 years of age, should everyone from the bottom quartile be moved to the second lowest quartile, was estimated to be 21.4% when comparing unrelated individuals in the population and 14.8% when comparing full siblings. This shift assumed that those with the lowest fitness would increase their fitness level (i.e., $W_{max}$) by 14% on average. However, whether such changes can be achieved through large-scale public health and community interventions, and whether the 14.8% observed in sibling comparisons will hold up under further causal scrutiny, remains to be explored. It seems plausible that this represents an upper bound, as several genetic, environmental, and behavioral confounders may differ even between full siblings. We want to underscore that the exact point estimates should be very cautiously interpreted given that PAF estimations also rely on strong assumptions. Nevertheless, comparisons of PAF estimates derived from standard cohort analysis versus those from sibling analysis may still be informative with respect to facilitating the clinical interpretation of the degree of influence that shared familial factors may play in the association between fitness and cancer.

In conclusion, this study suggests that high adolescent cardiorespiratory fitness is associated with lower overall cancer mortality even when comparing full siblings and thereby controlling for all behavioral, environmental, and genetic confounders that they share. However, the study also provides evidence suggesting that there might be varying confounding effects by specific cancer sites. For instance, the excess risks of prostate and non-melanoma skin cancer risks did not replicate when comparing full siblings, possibly due to screening biases. These findings suggest that epidemiological studies on cardiorespiratory fitness and cancer might yield biased conclusions if not conducted carefully, especially considering the bias appears to vary if one considers mortality or diagnosis, and that the associations with different cancer types may be biased to different degrees.

## Supporting information

**S1 Table. Diagnostic codes used to define the outcomes in the study.**
(DOCX)

**S2 Table. Baseline characteristics in in the total cohort and the full sibling cohort for those where covariate data was available and for those where covariate data was missing.**
(DOCX)

**S3 Table. Follow-up time, number of events, and numbers censored in cohort and sibling analysis.**
(DOCX)

**S4 Table. Follow-up time, cancer-specific deaths, and numbers censored for site-specific mortality in cohort and sibling analysis.**
(DOCX)

**S5 Table. Hazard ratios for cancer by quartiles of cardiorespiratory fitness in cohort and sibling analysis.**
(DOCX)

**S6 Table. Standardized cumulative incidence of cancer at 65 years of age by quartiles of cardiorespiratory fitness in cohort and sibling analysis.**
(DOCX)

**S7 Table. Population-attributable fraction for cancer at 65 years of age by cardiorespiratory fitness in cohort and sibling analysis, when considering a moderate (shifting those in quartile 1 to quartile 2) or a major intervention (shifting everyone to quartile 4).**
(DOCX)

**S8 Table. Hazard ratios for cancer by quartiles of cardiorespiratory fitness in cohort analysis, in the sibling cohort using standard analysis, and in the sibling cohort using sibling analysis.**
(DOCX)

**S9 Table. Hazard ratios for cancer by quartiles of cardiorespiratory fitness in cohort and sibling analysis, with and without adjusting for body mass index.**
(DOCX)

**S10 Table. Net risk and crude risk of cancer at 65 years of age by quartiles of cardiorespiratory fitness in cohort and sibling analysis.**
(DOCX)

**S11 Table. Hazard ratios for overall cancer diagnosis and mortality by quartiles of cardiorespiratory fitness in cohort analysis and sibling analysis using Cox regression compared to using Flexible parametric regression.**
(DOCX)

**S12 Table. Hazard ratios for overall cancer diagnosis and mortality by quartiles of cardiorespiratory fitness with and without allowing for effect modification by BMI in cohort and sibling analysis.**
(DOCX)

**S13 Table. Hazard ratios for overall cancer diagnosis and mortality by quartiles of cardiorespiratory fitness in cohort and sibling analysis using complete-case analysis versus using multiple imputation.**
(DOCX)

**S14 Table. Hazard ratios for overall cancer diagnosis and mortality by quartiles of cardiorespiratory fitness in cohort and sibling analysis using various numbers of knots for the baseline hazard based on Harrell's recommendations.**
(DOCX)

**S15 Table. Standardized cumulative incidences of cancer at age 65 by quartiles of cardiorespiratory fitness in cohort and sibling analysis using various numbers of knots for the baseline hazard based on Harrell's recommendations.**
(DOCX)

**S16 Table. Time-specific hazard ratios for cancer by quartiles of cardiorespiratory fitness in cohort and sibling analysis.**
(DOCX)

**S17 Table. Hazard ratios for site-specific cancer mortality by quartiles of cardiorespiratory fitness in cohort analysis.**
(DOCX)

**S1 Fig. Participant flow chart.**
(DOCX)

**S2 Fig. Directed acyclic graph for the association between adolescent cardiorespiratory fitness and risk of cancer in late adulthood, where observed (white) and unobserved (gray) confounders are illustrated.**
(DOCX)

**S3 Fig. Hazard ratios for overall cancer diagnosis and mortality and site-specific cancer (diagnosis or death) across restricted cubic splines of cardiorespiratory fitness in cohort (blue) and sibling analysis (red).**
(DOCX)

**S4 Fig. Standardized cumulative incidence for site-specific cancer (diagnosis or death) by quartiles of cardiorespiratory fitness in cohort and sibling analysis.**
(DOCX)

**S5 Fig. Standardized cumulative incidence for site-specific cancer (diagnosis or death) by quartiles of cardiorespiratory fitness in cohort and sibling analysis.**
(DOCX)

**S1 Checklist. RECORD checklist.**
(DOCX)

## Acknowledgments

S1 and S2 Figs were created with BioRender.com, see https://BioRender.com/zd6mzpp and https://BioRender.com/1y07w3x for CC-BY licensing.

## Author contributions

**Conceptualization:** Marcel Ballin, Daniel Berglind, Pontus Henriksson, Martin Neovius, Anna Nordström, Francisco B. Ortega, Elina Sillanpää, Peter Nordström.

**Data curation:** Peter Nordström, Viktor H. Ahlqvist.

**Formal analysis:** Marcel Ballin.

**Investigation:** Marcel Ballin, Daniel Berglind, Pontus Henriksson, Martin Neovius, Anna Nordström, Francisco B. Ortega, Elina Sillanpää, Peter Nordström, Viktor H. Ahlqvist.

**Methodology:** Marcel Ballin, Daniel Berglind, Pontus Henriksson, Martin Neovius, Anna Nordström, Francisco B. Ortega, Elina Sillanpää, Peter Nordström, Viktor H. Ahlqvist.

**Project administration:** Marcel Ballin, Viktor H. Ahlqvist.

**Resources:** Peter Nordström.

**Supervision:** Peter Nordström, Viktor H. Ahlqvist.

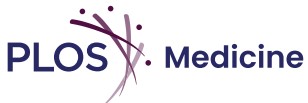

**Validation:** Viktor H. Ahlqvist.

**Visualization:** Marcel Ballin, Viktor H. Ahlqvist.

**Writing – original draft:** Marcel Ballin.

**Writing – review & editing:** Marcel Ballin, Daniel Berglind, Pontus Henriksson, Martin Neovius, Anna Nordström, Francisco B. Ortega, Elina Sillanpää, Peter Nordström, Viktor H. Ahlqvist.

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
