## [Editor Report · Decision Letter 0]

21 Oct 2024

Dear Dr Ballin, 

Thank you for submitting your manuscript entitled "Adolescent Cardiorespiratory Fitness and Risk of Cancer in Late Adulthood: A Nationwide Sibling-Controlled Cohort Study" for consideration by PLOS Medicine.

Your manuscript has now been evaluated by the PLOS Medicine editorial staff and I am writing to let you know that we would like to send your submission out for external peer review.

Please re-submit your manuscript within two working days, i.e. by Oct 23 2024.

Feel free to email me at atosun@plos.org or us at plosmedicine@plos.org if you have any queries relating to your submission.

Kind regards,

Alexandra Tosun, PhD

Associate Editor

PLOS Medicine

---

## [Decision Letter · Decision Letter 1]

17 Dec 2024

Dear Dr Ballin,

Many thanks for submitting your manuscript "Adolescent Cardiorespiratory Fitness and Risk of Cancer in Late Adulthood: A Nationwide Sibling-Controlled Cohort Study" (PMEDICINE-D-24-03504R1) to PLOS Medicine. The paper has been reviewed by subject experts and a statistician; their comments are included below and can also be accessed here: [LINK]

As you will see, the reviewers had mixed impressions of the manuscript and raised several points to improve clarity and avoid potential confusion. After discussing the paper with the editorial team and an academic editor with relevant expertise, I'm pleased to invite you to revise the paper in response to the reviewers' comments. We plan to send the revised paper to some or all of the original reviewers, and we cannot provide any guarantees at this stage regarding publication.

We ask that you submit your revision by Jan 07 2025. However, if this deadline is not feasible (including due to the upcoming holidays), please contact me by email and we can discuss a suitable alternative. Please note that the editorial team will be out of office from 23 December 2024 up to and including 3 January 2025.

Don't hesitate to contact me directly with any questions (atosun@plos.org). 

Best regards, 

Alexandra 

Alexandra Tosun, PhD 

Associate Editor

PLOS Medicine

atosun@plos.org

Comments from the reviewers: 

Reviewer #1: Using a Swedish population-level conscription database (and other linked datasets), this research investigated the relationship between adolescent cardiovascular fitness and time to subsequent cancer diagnoses and mortality, controlling for key, known confounders. Differentiating this research from earlier, similar research, the authors controlled for some unobserved confounding using between-within modelling of sibling clusters (in addition to modelling broader cohort) and flexible parametric survival models allowing estimation of baseline hazard instead of the more traditional Cox regression. 

Overall, this manuscript is an excellent initial submission. It is clearly written, logically presented, and cogently argued. The high level of care taken by the authors to use methods that produce more interpretable and rigorous results is laudable. Relatedly, the authors have recognised some of their methods are relatively novel and cited appropriate methodological work providing helpful background and justifying their choices well. That said, there are some areas of potential confusion and opportunities to improve clarity. 

Detailed feedback on the manuscript is provided below. My primary focus has been methodological, but I have made a few broader points as well. All items are major unless indicated otherwise. 

1. The authors prosecute a compelling argument against analysing composite endpoints of cancer detection and survival. They then use the precise approach they argued against in their specific cancer analyses. While they note this apparent contradiction in their discussion, it is not sufficiently explained or justified; noting, of course, insufficient sample size is no justification for poor analytical choices. Suggest the authors provide a compelling argument as to why they are going against their own recommendations or adjust their analyses to align e.g., by including analyses of cancer-specific survival only (and omitting detection); noting the authors similarly argue that cancer detection alone as an endpoint is misleading due to the potential confounding of cardiovascular fitness by health-seeking behaviours.

2. Methodological aspects of the authors' presentation of sibling comparison studies seems more optimistic than the literature they cite. For example, they authors say "One possible way to account for unobserved factors is to compare family members, such as siblings;[24] who are assumed to share a large part of these behaviours and other important genetic and environmental confounders.[25,26]". The methodological paper cited ([24]) however says "A popular way to reduce confounding bias is to use sibling comparisons, which implicitly adjust for several factors in the early environment or upbringing without requiring them to be measured or known." Indeed, the assumption that any given potential confounder is sufficiently shared between siblings to provide adequate control in any modelling is particularly important and perhaps not yet recognised sufficiently clearly in the manuscript. Suggest the authors consider moderating their language throughout the manuscript (including the abstract) to reflect that sibling comparison studies account for some, but not necessarily "most" or "all", of potential confounders and including key assumptions/limitations. 

3. Relatedly but separately, in the Methods/Statistical Analysis the authors say "For the full-sibling analysis, the aforementioned model was extended to a marginalized between-within model with robust standard errors, which enables control for all unobserved confounders (including genetic and shared environmental factors)." Saying all unobserved confounders are controlled is clearly not correct. Wording similar to that used elsewhere in the manuscript should be used here noting control of all unobserved confounders _shared_between_the_siblings_. 

4. [MINOR] In reference 43 the authors note that traditional Cox regression HR results are very similar to those obtained from their flexible parametric approach. Consider including this point as it potentially reassures readers more familiar with Cox regression than the flexible parametric approach employed. 

5. It's not clear to me how reference 45 relates to this research. It appears to focus on non-survival endpoints and the only reference to survival endpoints (last para of the discussion) refers to results from a separate paper using a fully parametric approach rather than the flexible parametric approach employed by the authors. Apologies if I am missing something here but would appreciate clarification. 

6. Did the authors consider analysing BMI as an effect modifier (i.e., including an interaction between BMI and CV fitness)? Either way, suggest providing some more detail on their decisions regarding how they chose to treat BMI once acknowledged as a "mediator". 

7. While data missingness leading to subject exclusion from the cohort was objectively low, it is interesting that missing covariate data caused more than double the number of patients to be excluded than missing endpoint data. Suggest the authors provide further information on what variables are driving missingness, whether this introduces any potential biases, and, if so, what was done to minimise the impact. This is particularly important given the small numbers in some of the specific cancer analyses. 

8. [MINOR] Ensure confidence intervals are reported consistently across the results (e.g., incidence difference). 

9. Suggest saying why smoking was not adjusted for in the analyses. I assume this is because a smoking variable was not available in any of the included datasets but this is worth saying explicitly given how important it is as a modifiable risk factor and, as noted by the authors, would have been particularly useful in the analysis of lung cancer specifically. 

10. Given the historical and ongoing underrepresentation of women as participants in clinical and observational health research, extreme care should be taken making assertions like this in the discussion "While we may believe there are limited reasons for sex-specific mechanisms linking cardiorespiratory fitness and physical activity to cancer …". At a minimum prior, highly reputable research should be cited and clear arguments made for taking this position. 

11. [MINOR] Notwithstanding the myriad problems associated with the use and interpretation of p-values, I suggest one is included with the point estimate and 95% CI in the very specific case where a result is explicitly said to be "statistically significant" e.g., in the discussion "The association between high adolescent cardiorespiratory fitness and lower risk of overall cancer mortality remained statistically significant…" 

12. [MINOR] The authors and the cited literature very effectively argue against the sole reliance on hazard ratios to characterise exposure-outcome relationships in survival analyses. Consider this advice with respect to the presented abstract. 

13. [MINOR] Given the relative novelty of some of the methods used, suggest including representative or actual Stata code. 

14. [MINOR] Ideally tables and figures should be interpretable independently of the main paper. Suggest removing references to "main article" or similar in supplement and, instead, providing sufficient context in the text accompanying the table or figure. 

Reviewer #2: This is an interesting study that attempts to examine the relationship between cardiorespiratory fitness measured in adolescence and later life risk of developing cancer. The data resource employed is novel and the analytic strategy for this type of data seems appropriate. The main challenge of the paper was the investigation of potential unmeasured confounding using sibling-oriented analysis while measuring few if any risk factors for cancer. 

Statements that the analysis works by "…controlling for all behavioural, environmental, and genetic confounders that they [siblings] share." is an extraordinarily strong statement with strong assumptions that cannot be tested in this study. 

It is reasonable to envision the divergence of many behavioral risk factors over 38 years of follow-up and the results of the study point to this as an issue in this study. 

Yet, the authors in this reviewer's reading work to interpret the results in both ways, (a) bias is a problem and (b) early life fitness is associated with lower risk for some cancers. 

It's a confusing message that does not make an important or adequately strong contribution for publication in a top-tier medical journal. 

Additional comments

In the context of risk factors that are not well established, and fitness and cancer risk certainly fall into this category, estimating PAF's may not be appropriate. 

The authors make no mention of logical biological mechanisms that provide a direct link between cardiorespiratory fitness (a phenotypic response to behavior and genetics) and cancer biology. The plausibility of any association observed is not well supported here. 

This report should find a good home somewhere in the literature, but not sure it's of sufficiently high quality for this journal—too many untestable assumptions. 

Reviewer #3: This study addresses a timely and important issue, and the authors have taken advantage of a very valuable data source in addressing this issue. A very comprehensive approach to data analysis was undertaken, and the findings provide an important advance in our understanding of the relationship between cardiorespiratory fitness (CRF) during adolescence and cancer risk later in life. The conclusions are appropriately supported by the findings. The manuscript is well organized and, for the most part, clearly written. However, the quality of the manuscript could be enhanced by addressing the following concerns:

1. The phrasing of the Objective in the Abstract is awkward. The phrase after the comma is not clear.

2. The purpose statement provided at the end of the Introduction is very long and convoluted. It would be more effective if it was split into two sentences.

3. In the section on measurement of CRF, the units for expression of Wmax should be provided. Was this expressed relative to body weight? Also, units should be provided in Table 1.

4. The first sentence in the Discussion section seems odd and unclear. This does not seem like an effective way to start the Discussion. While the point is significant, it does seem to be the major finding of the study.

5. In the Strengths and Limitations section, the authors have done a good job of noting most of the study's limitations. However, they fail to mention that, while the sibling analysis is helpful, the data source includes no information on the subjects' physical activity either at baseline or at any point during follow-up. 

Reviewer #4: It is my pleasure to review this work. The study investigated the associations between adolescent cardiorespiratory fitness and cancer outcomes later in life using a large sibling-controlled cohort of Swedish men who underwent military conscription examinations. Over a median follow-up of 55.9 years, the study found that higher adolescent fitness levels were associated with a reduced risk of overall cancer mortality, as well as decreased risks for several site-specific cancers such as rectum, head and neck, bladder, stomach, pancreas, colon, and others. However, higher risks were observed for prostate and skin cancers, leading to an increased overall cancer diagnosis risk.

Sibling comparisons revealed that shared familial confounders accounted for some of these associations, particularly the higher risks for prostate and skin cancers, which attenuated to null in this analysis. The findings underscore the protective effects of high adolescent cardiorespiratory fitness on cancer mortality but highlight the variable role of confounding across cancer types.

I have a few suggestions that may improve the manuscript.

1. Derivation of study population, the authors have excluded 72,042 individuals due to missing covariate data. The authors should provide the exclusion percentages for each covariate and compare the baseline characteristics between the entire study population and the population after excluding individuals with missing covariates. Moreover, the proportion of missing covariates exceeds 5% (72,042/1,249,131). The authors may consider the use of multiple imputation in the primary analysis and supplement the sensitivity analysis with results from the complete case analysis.

2. The authors adjusted for confounders including age at conscription, year of conscription, BMI, parental education, and parental income in the primary analysis. Could the authors clarify the criteria for adjusting for these confounders? Please also include a directed acyclic graph (DAG) to illustrate the relationships between exposure, outcome, and covariates.

3. The authors used flexible parametric survival models (FPSMs) to estimate hazard ratios. As far as I know, FPSMs rely on selecting an appropriate spline function form and the placement of knots to model the baseline hazard function. The authors selected the 5th, 27.5th, 50th, 72.5th, and 95th percentiles as knot positions. Could the authors clarify how these specific knot positions were chosen?

4. When modeling cardiorespiratory fitness using restricted cubic splines (RCS), the knot positions were selected at the 5th, 35th, 65th, and 95th percentiles. What was the rationale behind this selection? The knot positions here differ from those used in the previous analysis. Could the authors explain the relationship between these two sets of knot placements?

5. FPSMs offer advantages in flexible modeling and estimating absolute effect sizes. I am wondering whether the results from the Cox proportional hazards model are consistent with the current findings. If the results from both models are consistent, the conclusions of the study would be more robust. The authors may include the results from Cox regression in the sensitivity analysis.

6. Cardiopulmonary health is a reliable indicator of long-term physical activity, and meeting the physical activity levels recommended by the WHO guidelines has a positive effect on cancer mortality and incidence. Therefore, physical activity may be an important confounder in this study, but it seems that the authors did not adjust for it. Full siblings may have different physical activity patterns, how did the authors address this important confounder of physical activity in their analysis?

7. Among the 14 site-specific cancers in men, pancreatic cancer is likely to be strongly associated with alcohol consumption, while lung cancer is likely to be strongly associated with smoking behavior. I am not sure if I agree the statement "siblings may share smoking patterns" in the limitations section. Smoking and alcohol consumption may be two important confounders in this study, and although they may not be available in the database, are there any alternative variables for smoking and alcohol use, such as smoking disorders and alcohol use disorders?

8. The authors categorized adolescent cardiorespiratory fitness into quartiles. Could the authors clarify the rationale for this categorization? Are there alternative categorization schemes that might be more clinically or public health relevant?

9. Maternal factors such as singleton status, low birth weight, and preterm birth may affect adolescent cardiorespiratory fitness. Did the authors consider the influence of these maternal factors on the study outcomes?

10. In comparing Q4 to Q1 for adolescent cardiorespiratory fitness, the authors reported higher risks of prostate and skin cancer, and because these are the most common cancer types, this led to an overall higher risk of cancer diagnosis (HR 1.08; 1.06-1.11). I am also interested in the broader overall cancer diagnosis risk, such as cases with ICD-10 diagnosis codes C00-C97. Could the authors provide results for this broader cancer category?

11. It would be interesting to examine how early-life cardiorespiratory health influences cancer risk and progression over the long term. Could the authors provide the impact of cardiorespiratory fitness on cancer outcomes at different time lags (e.g., 10 years, 20 years)?

12. Please include the information on disease incidence density in the results section.

13. Please add a description of the follow-up start and end points for the siblings.

14. Please ensure that the number of decimal places for incidence differences is consistent, for example, 1.58 (1.0 to 1.9) percentage point difference and 0.08 (-0.9 to 1.1).

---

* Please upload any figures associated with your paper as individual TIF or EPS files with 300dpi resolution at resubmission; please read our figure guidelines for more information on our requirements: http://journals.plos.org/plosmedicine/s/figures. While revising your submission, please upload your figure files to the PACE digital diagnostic tool, https://pacev2.apexcovantage.com/. PACE helps ensure that figures meet PLOS requirements. To use PACE, you must first register as a user. Then, login and navigate to the UPLOAD tab, where you will find detailed instructions on how to use the tool. If you encounter any issues or have any questions when using PACE, please email us at PLOSMedicine@plos.org.

* FINANCIAL DISCLOSURES: The funding statement should include: specific grant numbers, initials of authors who received each award, URLs to sponsors’ websites. Also, please state whether any sponsors or funders (other than the named authors) played any role in study design, data collection and analysis, the decision to publish, or preparation of the manuscript. If they had no role in the research, include this sentence: “The funders had no role in study design, data collection and analysis, decision to publish, or preparation of the manuscript.”

* COMPETING INTERESTS: All authors must declare their relevant competing interests per the PLOS policy, which can be seen here: https://journals.plos.org/plosmedicine/s/competing-interests

For authors with ties to industry, please indicate whether any of the interests has a financial stake in the results of the current study.

FIGURES AND TABLES

SUPPLEMENTARY MATERIAL

REFERENCES

* Where website addresses are cited, please include the complete URL and specify the date of access (e.g. [accessed: 12/06/2024]).

STUDY TYPE-SPECIFIC REQUESTS

* Abstract: Please include the study design, population and setting, number of participants, years during which the study took place (enrollment and follow up), length of follow up, and main outcome measures.

* Please ensure that the study is reported according to the RECORD guideline (available from https://www.record-statement.org) and include the completed checklist as Supporting Information. Please add the following statement, or similar, to the Methods: "This study is reported as per the Reporting of Studies Conducted using Observational Routinely-Collected Data (RECORD) guideline (S1 Checklist)." When completing the checklist, please use section and paragraph numbers, rather than page numbers.

* For all observational studies, in the manuscript text, please indicate: (1) the specific hypotheses you intended to test, (2) the analytical methods by which you planned to test them, (3) the analyses you actually performed, and (4) when reported analyses differ from those that were planned, transparent explanations for differences that affect the reliability of the study's results. If a reported analysis was performed based on an interesting but unanticipated pattern in the data, please be clear that the analysis was data driven. 

* Please state in the Methods section whether the study had a prospective protocol or analysis plan. If a prospective analysis plan (from your funding proposal, IRB or other ethics committee submission, study protocol, or other planning document written before analyzing the data) was used in designing the study, please include the relevant document(s) with your revised manuscript as a Supporting Information file to be published alongside your study and cite it in the Methods section. A legend for this file should be included at the end of your manuscript. If no such document exists, please make sure that the Methods section transparently describes when analyses were planned, and when/why any data-driven changes to analyses took place. Changes in the analysis, including those made in response to peer review comments, should be identified as such in the Methods section of the paper, with rationale.

---

## [Decision Letter · Decision Letter 2]

21 Feb 2025

Dear Dr Ballin,

Many thanks for re-submitting your manuscript "Adolescent Cardiorespiratory Fitness and Risk of Cancer in Late Adulthood: A Nationwide Sibling-Controlled Cohort Study" (PMEDICINE-D-24-03504R2) to PLOS Medicine. The paper has been seen again by subject experts and a statistician; their comments are included below and can also be accessed here: [LINK]

Thank you for your detailed response to the reviewers' comments. As you will see, there is mixed feedback from the reviewers, some of which requires further clarification. We have discussed the manuscript within the editorial team and with the academic editor and have decided to consider the manuscript further. Please note that for some of Reviewer 4's comments we have consulted with the statistical reviewer who has provided additional input. You will also find additional input from the editorial team that we ask you to consider in your revision. After discussing the paper with the editorial team, we ask you to carefully address the comments in a further revision. We plan to send the revised paper to some or all of the original reviewers.

We ask that you submit your revision by Mar 14 2025. However, if this deadline is not feasible, please contact me by email, and we can discuss a suitable alternative.

Don't hesitate to contact me directly with any questions (atosun@plos.org). 

Best regards, 

Alexandra 

Alexandra Tosun, PhD 

Associate Editor

PLOS Medicine

atosun@plos.org

Comments from the editorial team:

Please look carefully at the comments provided by the reviewers and the additional input provided by the statistical reviewer. We feel that it is very important to balance and tone down the presentation of conclusions based on potential confounding. We agree with reviewer #4 regarding their comments on physical activity and feel that physical activity should be considered as a confounder. If the data are not available, this should be discussed more clearly as a strong limitation.

Comments from the reviewers: 

Reviewer #1: Thanks to the authors for their considered responses to my queries. Their clarifications have addressed all my feedback, and the final version of the manuscript is notably strong in how the methodological details are presented. Please accept my congratulations for an excellent paper.

Reviewer #2: I continue to have limited enthusiasm for this report. 

Reviewer 1 and I largely agree on the authors' overly strong assumptions related to confounding problems caused by common cancer risk factors that were not directly examined in this report. Indeed, the authors state in response to my first comment that, "While these assumptions are strong and untestable, they are inherent to the methodology and cannot be empirically verified, even with extensive data collection." It not uncommon for many siblings to diverge in their life course regarding key cancer risk factors such as weight gain, alcohol consumption, smoking, and exercise, which raises questions about the effectiveness of the method used to control for these factors in this analysis. 

In response to my second comment that was also focused on this issue, the authors chose to focus their response on the stability fitness over time vs. divergence of risk factors in siblings that could bias their results.

The evidence linking fitness to incident cancer risk remains somewhat limited, and the studies cited (refs 5-9) do not provide strong insight. Refs 5 and 9 appear to be reviews of the same studies, while the study with better control over relevant cancer risk factors (ref 8) found an association with only colon and lung cancer. Furthermore, the latter cancer site is often confounded by smoking due to the strong associations between smoking and lung cancer (i.e., residual confounding resulting from measurement error). Many believe that estimates of PAF assume that risk factors have been convincingly linked to the outcome of interest, and using this metric to describe the public health impact of changes in behaviors with more limited evidence is not advisable. 

Last, the authors provided no plausible mechanism linking fitness (not activity) to biological pathways that may reduce cancer risk. They seem to suggest that activity increases fitness, and the mechanisms related to activity may also be operative for fitness. Perhaps providing evidence of mechanisms linking fitness to cancer in the context of a causal diagram that provides evidence for the direct effects of fitness on biological mechanisms related to cancer independent of physical activity would be useful. If the mechanisms are activated by exercise/activity and fitness is just a bystander in relation to cancer risk, what does that say about the potential for a putative causal effect of fitness on cancer risk?

Reviewer #4: Although the authors have addressed most of the reviewer's concerns and presented and explained the relevant results in the manuscript, there are certain points in the response letter with which we disagree. We still have the following issues:

1. For question #1: The authors have addressed our concerns and supplemented the results with those from multiple imputation, which are consistent with the results from complete case analysis. However, the authors suggest that most of the missing data are based on mechanisms of missing completely at random (MCAR) and missing not at random (MNAR). Furthermore, they state that in the case of missing data being missing at random (MAR), multiple imputation would lead to less bias compared to complete case analysis. I am puzzled by the authors' assertion that "most of the missing data are based on mechanisms of MCAR and MNAR." My main concern is: what is the rationale behind the authors' speculation that "most of the missing data are based on MCAR and MNAR mechanisms"?

In the supplementary analysis (Table S2), it is clear that the median year of birth for individuals with complete covariate data is 1966 (1960 to 1971), while for those excluded, the median year of birth is 1956 (1947 to 1977). The difference of 10 years in the median birth year between these two groups raises the question: does the author believe there is no significant difference? This is just an example to illustrate that some covariates may have substantial differences between the groups. This highlights the point that the missing data mechanism—whether MNAR or MAR—cannot be inferred from observed data alone (BMJ 2009;338:b2393).

2. For question #3: The authors used flexible parametric survival models (FPSMs) to estimate hazard ratios. However, they argue that this approach is important for making the findings clinically and publicly relevant, as hazard ratios alone are difficult to interpret. This seems somewhat contradictory. Why is it that hazard ratios alone are difficult to interpret? I am also confused about the benefit of estimating the standardized failure function or cumulative incidence using FPSMs for public health significance. Since the authors believe that hazard ratios alone are hard to interpret, for example, shouldn't they have used restricted mean survival time (RMST) to make the estimates instead? (e.g., Multiply robust estimator for the difference in survival functions using pseudo-observations. BMC Medical Research Methodology. 2023. PMID: 37872495)

3. For question #6: We have significant disagreements with the authors regarding the issue of whether physical activity should be considered an important confounder in this study. The authors are uncertain whether physical activity has a (non-trivial) causal effect on cancer risk independent of fitness. However, numerous high-quality observational studies and Mendelian randomization studies have shown that both self-reported and accelerometer-measured physical activity can significantly reduce cancer risk. These studies provide strong evidence supporting physical activity as an independent protective factor. Therefore, we believe that viewing physical activity as solely influencing cancer risk through fitness levels is insufficient.

From both of these perspectives, physical activity not only reduces cancer risk through improving fitness but also acts as an independent exposure factor that likely affects cancer risk through multiple mechanisms. We therefore consider physical activity to be an important confounder in cancer risk.

While we understand that the study was limited by the lack of data on physical activity, which should be addressed in the limitations section. However, we do not agree with the authors' neglect of physical activity as a potential confounder.

Additional comments from the statistical reviewer: 

Point 1: The reviewer seems to accept the newly provided results that complement the full case analysis with MI for the relatively small amount of missing data. They seem to have concerns about the reasoning behind the missingness mechanisms proposed by the author. While I agree that it would be interesting to hear more of the authors' reasoning for why they think the data are MCAR and MNAR, it doesn't ultimately make any difference to the analyses underlying the current research for two reasons. Firstly, the amount of missing data is very small, so the likelihood of the missing data having a significant impact on the conclusions of this research is minimal. Secondly, and with this in mind, the authors have already presented results using CCA and MI to deal with missingness and have shown comparable results. MI is valid for MCAR and MAR, but may be biased for MNAR if CCA should be considered (https://academic.oup.com/ije/article/48/4/1294/5382162). By presenting both CCA and MI results, the authors have adequately covered their bases.

Point 2: The reviewer makes a similar point to my own review point:

"""12. [MINOR] The authors and the cited literature very effectively argue against the sole reliance on hazard ratios to characterise exposure-outcome relationships in survival analyses. Consider this advice with respect to the presented abstract.

Response: Thank you for this suggestion. We have carefully revised the abstract to include absolute measures alongside the hazard ratios."""

The interpretability of hazard ratios is a well-documented challenge, but they are also the main statistic for survival/time to event analyses and this is not likely to change in the near future. With this reality in mind, I think the authors have done a good job of walking the fine line between raising the issues with hazard ratios, but also presenting them for comparability with other research in addition to their absolute effect estimates made available by using FPSMs, i.e. "These [FSPM] models yield similar HRs as conventional Cox regression, while having the additional advantage of directly estimating the baseline hazard, thereby enabling the computation of absolute effects measures" (see lines 234 to 237 in the latest submission). I think this fully addresses the reviewer's point and obviates the need for other statistics such as RMST.

---

## [Decision Letter · Decision Letter 3]

1 Apr 2025

Dear Dr. Ballin,

Thank you very much for re-submitting your manuscript "Adolescent Cardiorespiratory Fitness and Risk of Cancer in Late Adulthood: A Nationwide Sibling-Controlled Cohort Study" (PMEDICINE-D-24-03504R3) for review by PLOS Medicine.

Thank you for your detailed response to the reviewers' comments. I have discussed the paper with my colleagues and the academic editor, and it has also been seen again by two of the original reviewers. The changes made to the paper were satisfactory to the reviewers. As such, we intend to accept the paper for publication, pending your attention to the editors' comments below in a further revision. When submitting your revised paper, please once again include a detailed point-by-point response to the editorial comments.

[LINK]

In revising the manuscript for further consideration here, please ensure you address the specific points made by the editors. In your rebuttal letter you should indicate your response to the reviewers' and editors' comments and the changes you have made in the manuscript. Please submit a clean version of the paper as the main article file. A version with changes marked must also be uploaded as a marked up manuscript file. Please also check the guidelines for revised papers at http://journals.plos.org/plosmedicine/s/revising-your-manuscript for any that apply to your paper.

We ask that you submit your revision within 1 week (Apr 08 2025). However, if this deadline is not feasible, please contact me by email, and we can discuss a suitable alternative.

Please do not hesitate to contact me directly with any questions (atosun@plos.org). If you reply directly to this message, please be sure to 'Reply All' so your message comes directly to my inbox.

We look forward to receiving the revised manuscript.

Sincerely,

Alexandra Tosun, PhD

Associate Editor

PLOS Medicine

plosmedicine.org

Comments from Reviewers:

Reviewer #1: Thank you for the opportunity to review the ongoing discussion regarding this manuscript. I remain supportive of its acceptance for publication. 

Reviewer #4: The authors have effectively addressed our initial concerns and made appropriate revisions to the paper. Congratulations on the great work.

[LINK]

Requests from Editors:

GENERAL

* Please ensure that all abbreviations are defined at first use throughout the text (including statistical abbreviations). Please also check figures and tables.

* Please review your text for claims of novelty or primacy (e.g. 'for the first time', ‘novel’) and remove this language.

* Please check that any use of statistical terms (such as trend or significant) are supported by the data, and if not please remove them.

* Please ensure that tables and figures, including those in supplementary files, are appropriately referenced in the main text.

* Statistical reporting: Please revise throughout the manuscript, including tables and figures (including Supplementary Materials).

a) Please report statistical information as follows to improve clarity for the reader "22% (95% CI [13%,28%]; p</=)". 

b) Please separate upper and lower bounds with commas instead of hyphens as the latter can be confused with reporting of negative values. 

c) Please define statistical definitions at first use and repeat the abbreviated definitions (HR, CI etc.) for each set of parentheses.

* The link in the data availability statement leads to a page that says "The page could not be found". Please replace the link and be sure to provide a link that provides a clear point of contact for researchers interested in the data.

* Financial Disclosure: Please ensure that the details provided on page 29 are included in the relevant statement in the metadata in the online submission form.

* Competing Interest: Please ensure that the details provided on page 29 are included in the relevant statement in the metadata in the online submission form.

* Title: We suggest adding 'in Sweden' (or similar) to the title.

ABSTRACT

* Please confirm that your abstract complies with our requirements, including providing all the information relevant to this study type https://journals.plos.org/plosmedicine/s/submission-guidelines#loc-abstract

* l.54ff: Please clarify what "they" refers to (the highest quartile of fitness?). For clarity, we suggest changing the text in the preceding lines to: "In the cohort analysis, individuals in the highest quartile of fitness...".

* Please ensure that all numbers presented in the abstract are present and identical to numbers presented in the main manuscript text.

* l.60ff: We suggest changing to: Individuals in the highest quartile of fitness had a higher risk of overall cancer diagnosis (HR 1.08; 1.06 to 1.11, P<0.001, incidence difference 1.32; 0.94 to 1.70 percentage points), results driven by prostate and skin cancer being the most common types of cancer.

* l.64: We suggest removing the word "all" and simply stating "thereby controlling for unobserved shared confounders."

* In the Abstract Conclusion, first sentence, please make sure it is clear that the association refers to overall cancer mortality in late adulthood.

AUTHOR SUMMARY

* We suggest re-phrasing the bullet points under ‘Why Was This Study Done?’:

1) High cardiorespiratory fitness, a measure of regular physical activity and linked to genetics, has been associated with reduced cancer risk.

2) Improving fitness in young people is a growing public health focus.

3) Existing studies may have limitations, such as not fully accounting for other factors, such as participation in cancer screening and other health behaviors that may be more common in individuals with higher fitness levels.

* Please change ‘The researchers’ and ‘They’ to ‘We’.

* Please specify ‘behaviors’.

* Please clarify that you are referring to cardiorespiratory fitness on line 103.

* In the final bullet point of 'What Do These Findings Mean?', please include the main limitations of the study in non-technical language.

INTRODUCTION

* l.122: Similar to the Author Summary, we feel that the statement that cardiorespiratory fitness is a robust marker reflecting genetic predisposition, might be misleading. We suggest re-phrasing to make clear that genetic factors may influence cardiorespiratory fitness.

* l.151: We suggest removing "all" because it is unlikely that you can guarantee that you have in fact covered all previous studies.

* l.152: Are you 100% sure there is only one? We suggest adding a qualifier, such as "to the best of our knowledge".

METHODS AND RESULTS 

* If you have not already, please provide an explanation why liver, bile ducts, and gallbladder cancer were grouped as one.

* l.342ff: For the full sibling analysis, please ensure that you discuss all site-specific cancers for which you observed attenuation. We suggest removing the word ‘tendencies’ as you clearly observed attenuation. We also noted that you did not discuss the results for myeloma at all.

* l.360: “In contrast, associations with colon and head and neck cancer seemed more robust.” – please explain in more detail. It seems that for head and neck cancer the statement is not true. Please revise.

* Table 1: Is it correct that the cardiorespiratory fitness and Wmax values by quartile are exactly the same between the total cohort and the full sibling cohort?

* Figure 1 and 2: Please define ‘CI’.

* Figure S2: Figures cannot be reproduced from other sources that are not CC-BY. Please provide a different figure.

DISCUSSION

* As a general reminder, please be sure to acknowledge any limitations the reviewers have pointed out, such as lack of insight into biological mechanisms. Also, please ensure that the language throughout the Discussion is measured, i.e. that you do not overstate any of the findings and results, and that you interpret the results with caution based on the limitations.

* Please remove any subheadings in the Discussion, including the Conclusion subheading.

* l.437: Please explain what you mean by heterogeneous pattern. We believe that sibling analysis, with a few exceptions, mostly leads to attenuation of the results, i.e. we are not sure that heterogeneous pattern is an appropriate description here.

* ll.438-441: We do not fully agree that the results for colorectal cancer and head and neck cancer appear to be more robust. Please see our earlier comment and revise accordingly.

General Editorial Requests

---

## [Editor Report · Decision Letter 4]

3 Apr 2025

Dear Dr Ballin, 

On behalf of my colleagues and the Academic Editor, Andre P Kengne, I am pleased to inform you that we have agreed to publish your manuscript "Adolescent Cardiorespiratory Fitness and Risk of Cancer in Late Adulthood: A Nationwide Sibling-Controlled Cohort Study in Sweden" (PMEDICINE-D-24-03504R4) in PLOS Medicine.

I appreciate your thorough responses to the reviewers' and editors' comments throughout the editorial process. We look forward to publishing your manuscript, and editorially there are only a few remaining points that should be addressed prior to publication. We will carefully check whether the changes have been made. If you have any questions or concerns regarding these final requests, please feel free to contact me at atosun@plos.org.

Please see below the minor points that we request you respond to:

1) Please update the Financial Disclosure and Competing Interest statement in metadata in the online submission form with the information provided under the Acknowledgements section (Conflict of Interest Disclosures and Funding). Please note that names should only be initials of authors.

2) Thank you for expanding the results section to discuss all site-specific cancers. Please consider removing some of the numerical results (with confidence intervals, etc.) in the main text if the corresponding tables and figures are clearly referenced and the numerical results can also be found there. We believe this would simplify the text.

3) Please remove the ‘Implications’ subheading from the Discussion section.

Before your manuscript can be formally accepted you will need to complete some formatting changes, which you will receive in a follow up email (including the editorial points above). Please be aware that it may take several days for you to receive this email; during this time no action is required by you. Once you have received these formatting requests, please note that your manuscript will not be scheduled for publication until you have made the required changes.

PRESS

Sincerely, 

Alexandra Tosun, PhD 

Associate Editor 

PLOS Medicine